# Phase Changes in the Surface Layer of Stainless Steel Annealed at a Temperature of 550 °C

**DOI:** 10.3390/ma15248871

**Published:** 2022-12-12

**Authors:** Anna Sedláčková, Tatiana Ivanova, Miroslav Mashlan, Hana Doláková

**Affiliations:** 1Faculty of Science, Palacký University, 17. Listopadu 1192/12, 77900 Olomouc, Czech Republic; 2Science and Technology Park, Palacký University, Šlechtitelů 21, 78371 Olomouc, Czech Republic

**Keywords:** selective laser melting, stainless steel (1.4404), austenitic phase, ferritic phase, Mössbauer spectroscopy, scanning electron microscope, powder X-ray diffraction, energy-dispersive X-ray spectroscopy

## Abstract

Stainless steels have the advantage of forming a protective surface layer to prevent corrosion. This layer results from phase and structural changes on the steel surface. Stainless steel samples (1.4404, 316L), whose alloying elements include Cr, Ni, Mo, and Mn, were subjected to the study of the surface layer. Prism-shaped samples (25 × 25 × 3) mm^3^ were made from CL20ES stainless steel powder, using selective laser melting. After sandblasting with corundum powder and annealing at 550 °C for different periods of time (2, 4, 8, 16, 32, 64, 128 h), samples were studied by conversion X-ray Mössbauer spectroscopy (CXMS), conversion electron Mössbauer spectroscopy (CEMS), scanning electron microscopy (SEM), energy-dispersive X-ray spectroscopy (EDS), and X-ray diffraction (XRD). The main topics of the research were surface morphology and elemental and phase composition. The annealing of stainless steel samples resulted in a new surface layer comprising leaf-shaped crystals made of chromium oxide. The crystals grew, and their number increased as annealing time was extended. The amount of chromium increased in the surface layer at the expense of iron and nickel, and the longer the annealing time was set, the more chromium was observed in the surface layer. Iron compounds (BCC iron, mixed Fe–Cr oxide) were found in the surface layer, in addition to chromium oxide. BCC iron appeared only after annealing for at least 4 h, which is the initial time of austenitic–ferritic transformation. Mixed Fe–Cr oxide was observed in all annealed samples. All phase changes were observed in the surface layer at approximately 0.6 µm depth.

## 1. Introduction

Most high-temperature alloys, such as stainless steel, have a protective layer on their surface made of Cr_2_O_3_ due to the high affinity of chromium for oxygen. Obtaining a continuous and well-protecting layer of chromium oxide is limited by at least a 20% chromium amount [1]. In [1], the authors studied different steels, including 316L stainless steel. In this research, the oxidation of 316L steel samples was carried out in air at 900 °C in 20-h cycles. As a result of X-ray diffraction (XRD) measurements, they found different oxides, such as Fe_2_O_3_, (Fe_0.6_Cr_0.4_)_2_O_3_, and NiFe_2_O_4_. However, these samples were made by compacting them at a high pressure and then sintering in a vacuum furnace. Other studies, such as [2,3], discuss laser-melted samples. When laser-melted 316L stainless steel is annealed for a longer period at high temperatures, it relieves residual stresses, but it can also cause recrystallization, grain growth, and “austenite–ferrite” phase transformation [2]. Research on the effects of annealing 316L steel was also carried out at other temperatures and in different atmospheres, for example, in [3,4,5]. In [2], the authors observed a phase transformation of austenite to ferrite when annealing in an argon atmosphere at temperatures higher than 1100 °C. However, this change was not observed in [3], even though the annealing conditions were identical, which the authors explain by the cooling rate. On the other hand, in [4], the austenite–ferrite phase transformation was already observed at temperatures at approximately 400 °C, with a comparable annealing time. However, the authors do not mention in what atmosphere the annealing was carried out. Other publications discuss the dependence of the austenite–ferrite phase transformation on the Cr/Ni ratio [6,7,8,9] and the existence of dislocations created during the selective laser melting (SLM) process [10,11]. In most publications, the phase transformation is observed by XRD [1,2,3,4,5] or metallographic analysis [1,2,3]. Structural and phase transformations of a stainless steel powder for SLM were studied in [12]. In [13,14,15], the effect of annealing on the microstructure (metallographic analysis) and mechanical behaviors of parts made of 316L steel by the SLM was investigated. In [5], Mössbauer spectroscopy was used to study phase changes in iron-containing steels.

At the same time, Mössbauer spectroscopy [16] is an excellent non-destructive method for studying the structure and phase composition of iron-containing alloys and steels. The scattering geometry of the implementation of measurements using the detection of secondary X-ray radiation and conversion electrons allows for the bulk and surface of the samples to be studied, respectively. Mössbauer spectroscopy, similar to X-ray diffraction, does not require additional treatment of the samples; therefore, by using the conversion electron Mössbauer spectroscopy method, we study the surface of the samples without being affected by any post-processing treatments.

This work aims to study morphology and phase changes in the surface layer of stainless steel (316L) samples made by selective laser melting (SLM) from CL20ES stainless steel powder and annealed at 550 °C for different time periods. For the surface morphology study, scanning electron microscopy (SEM) was used. Phase change research involved X-ray diffraction (XRD), energy-dispersive X-ray spectroscopy (EDS), conversion electron Mössbauer spectroscopy (CEMS), and conversion X-ray Mössbauer spectroscopy (CXMS).

## 2. Materials and Methods

### 2.1. Sample Preparation

Prism-shaped samples (25 × 25 × 3 mm^3^) from CL20ES steel powder (chemical composition in Table 1) were manufactured using selective laser melting (Concept Laser M2-cusing system, GE Additive, Cincinnati, OH, USA). Yb:YAG diode-pumped fiber optical laser (wavelength of 1070 nm) with 200 W power was used for selective laser melting. The samples were sandblasted (corundum powder) after SLM manufacturing.

The samples were annealed in a laboratory furnace (LE05/11, LAC, Zidlochovice, Czech Republic) at a temperature of 550 °C in air. The samples were placed in the furnace at room temperature; the heating time to 550 °C was 1 h. The annealing lasted for 2, 4, 8, 16, 32, 64, 128 h; after the annealing time, samples spontaneously cooled to room temperature in the furnace.

### 2.2. Mössbauer Spectroscopy

The backscattering ^57^Fe Mössbauer spectra were accumulated by a Mössbauer spectrometer operating in constant acceleration mode and equipped with a ^57^Co(Rh) source and MS96 Mössbauer spectrometer software [17] at room temperature. Spectra were recorded on 512 channels. A proportional gas detector registering 6.4 keV X-rays was used to accumulate conversion X-ray Mössbauer spectra (CXMS). Conversion electron Mössbauer spectra (CEMS) were acquired with an air scintillation detector [18]. The least squares fit of the lines using the MossWinn 4.0 software program [19,20] performed the calculation and evaluation of the Mössbauer spectra. The isomer shift values referred to the centroid of the spectrum recorded from an α-Fe foil (thickness 30 µm) at room temperature.

### 2.3. X-ray Diffraction

X-ray measurements were carried out on an X-ray diffractometer Bruker Advance D8 (Bruker, Billerica, MA, USA). The unit is equipped with a Co K_𝛼_ X-ray tube and LYNXEYE position sensitive detector. Measurements were performed with Bragg–Brentano parafocusing geometry. The measurements were conducted at 35 kV for X-ray tube voltage and 40 mA for X-ray tube current. Experiments were performed with the step size 0.03°. The instrument was equipped with a 0.6 mm divergence slit and 2.5° axial Soller slits for the primary beam path and 20 µm Fe K_β_ filter and 2.5° axial Soller slits for the secondary beam path.

### 2.4. Scanning Electron Microscopy and Energy-Dispersive X-ray Spectroscopy

The VEGA3 LMU (TESCAN, Brno, Czech Republic) scanning electron microscope was chosen to study the surface morphology and elemental composition using energy-dispersive X-ray spectroscopy. The electron source in this microscope is the LaB_6_ cathode. The microscope includes two detectors: a secondary electron detector of the Everhart–Thornley type (TESCAN, Brno, Czech Republic) and an XFlash silicon drift detector 410-M (Bruker Nano GmbH, Berlin, Germany).

## 3. Results

### 3.1. Scanning Electron Microscopy

Figure 1 shows electron microscope images. Magnifications of 1000×, 10,000×, and 50,000× were selected for all images. Figure 1 (top) shows images of an unannealed sample and surface morphology of a steel part manufactured by SLM and sandblasted with corundum powder. No original particles remained on the surface due to sandblasting, as discussed in [5]. However, the surface was still not smooth but showed some roughness. The following images show the surface of the annealed samples. SEM images show that leaf-shaped crystals formed on the surface due to annealing. The shape of these crystals can be described as prisms with two dimensions much larger than the last dimension. As the annealing time increased, the number and size of these formations increased, but they kept their original shape. The increasing number of crystals can be seen on images with 10,000× magnification (Figure 1 middle). The leaf shape of these formations was evident at the highest magnification (Figure 1 right).

### 3.2. Energy-Dispersive X-ray Spectroscopy

EDS was used to determine the elemental composition of the surface layers of samples, including in-depth analysis. The elemental composition was monitored to observe the surface diffusion of the elements due to annealing. The acceleration voltages were 11, 13, 15, 20, 25, and 30 kV, which, according to Castaing’s formula [21], allows for obtaining information from depths of 0.3, 0.5, 0.7, 1.3, 1.9, and 2.6 μm, respectively. The results of the in-depth analysis are demonstrated in Figure 2. The amount of chromium increased with the expense of iron and nickel. The other elements (Mn and Mo) kept the original amount regardless of voltage. The amount of chromium increased depending on the acceleration voltage and annealing time. With increasing annealing time, the Cr/Fe ratio gradually increased in the surface layer by approximately 1 μm. The amount of elements in Figure 2 are in normalized wt% (100% corresponds to the sum of Fe, Cr, Ni, Mn, Mo). The inhomogeneity of the surface was shown by SEM images (Figure 1), and therefore, an EDS analysis was performed at selected points (Figure 3).

We should see a minimum in the amounts, for example, chromium, because the amount diffused to the surface should be missing in depth. This cannot be seen in Figure 2, which was caused by measuring at different depths. When the samples were measured at 11 kV, information was gained from 0.3 μm depth; when they were measured at 13 kV, the result was an average of information from 0.3 μm and other 0.2 μm under the first depth. Thus, when it comes to depth, where there is less chromium than in the original steel, we cannot see it, because it is only the average information. This also means that the thickness of the surface layer is much smaller than 1 μm.

Figure 3 shows two SEM images representing the sample’s surface annealed for 32 h. At an accelerating voltage of 15 kV, corresponding to a depth of 0.7 μm, elemental analysis was carried out at four points on the sample’s surface. These points are marked in yellow in the figure. The figure also shows the normalized percentages (wt% with an uncertainty of 3 wt%) of five selected elements occurring in the original steel in at least 1% quantity.

The measurement at point 1 corresponds to an almost empty surface without particles, while point 2 represents the elemental composition of the crystal on the surface. In this case, the amount of iron has decreased at the expense of chromium. This would mean that the crystals on the sample’s surface are mainly chromic. This supposition is confirmed by measuring at two other points, points 3 and 4. At these two points, there are large crystals, which means that when measuring with the same voltage, i.e., at the same depth, mainly the elemental composition of the crystals is measured. On the contrary, at point 2, the surface underneath the crystal is partly measured. Due to EDS measurements, it is apparent that crystals contain mainly chromium. The annealing was performed in air, which leads to an assumption that these crystals are made of chromium oxide.

### 3.3. Mössbauer Spectroscopy

Mössbauer spectra were accumulated by CEMS and CXMS. The reason for measuring with both methods was the possibility of comparing information obtained from different depths since the penetration depth of electrons is approximately 0.3 µm (CEMS), while X-rays penetrate up to a depth of 10 µm (CXMS).

#### 3.3.1. Conversion X-ray Mössbauer Spectroscopy

The CXMS spectra of all samples were practically the same and consisted of the broad line of FCC austenite. The accumulation time of one spectrum was usually approximately 40 h (^57^Co(Rh) activity approx. 5 mCi). These spectra were fitted with a doublet with minor quadrupole splitting. This model corresponds to iron atoms arranged in an FCC lattice, with the presence of alloying elements responsible for the existence of quadrupole splitting [22]. Only the sample spectrum annealed for 16 h was accumulated for 430 h. In this case, a very weak sextet (BCC ferrite) appeared in the spectrum, corresponding to iron atoms in the BCC crystal lattice. Spectrum of this sample can be seen in Figure 4. The sextet comprised only 5% of the spectrum (Figure 4 detail). A summary of the Mössbauer parameters is shown in Table 2; all spectra are in Figure 5.

The size of the Mössbauer effect was also investigated. As annealing time increased, a smaller effect was observed (Figure 6). The effect was calculated as the maximum of fit. This decreasing tendency is caused by the depth growth of the surface layer without iron blocking the ^57^Fe Mössbauer effect.

#### 3.3.2. Conversion Electron Mössbauer Spectroscopy

The CEMS spectra (Figure 7) of the annealed samples differed from the CXMS spectra. CEMS spectra contained a broad line corresponding to FCC austenite in all cases. Moreover, the CEMS spectra also contained a doublet, corresponding to trivalent iron per the value of the isomer shift. A sextet corresponding to BCC ferrite appeared after annealing for more than 4 h. The Mössbauer parameters of the spectra are listed in Table 3. When fitting with the MOSSWIN program, the parameters of the Fe^3+^ doublet were fixed for samples with a low Fe^3+^ doublet content. The fixed parameter values were from the CEMS spectrum of the sample annealed at 1000 °C for 2 h, which contained only the Fe^3+^ doublet.

### 3.4. X-ray Diffraction

Figure 8 shows the results of X-ray measurements. The austenitic iron phase (FCC phase) was present in all X-ray patterns. With an annealing time of ≥4 h, there was an iron ferritic phase (BCC phase). All annealed samples were characterized by mixed Fe–Cr oxide in the phase composition. With an increasing annealing time (≥8 h), chromium oxide Cr_2_O_3_ appeared on the surface of the samples.

## 4. Discussion

The samples were subjected to several methods to obtain as much information as possible about the structure and composition to compare obtained data and to confirm correctness.

The morphology study confirmed the presence of a new surface layer on all annealed samples. This layer comprised leaf-shaped crystals (Figure 1) containing chromium oxide Cr_2_O_3_ (Figure 3 and Figure 8). The grown crystals had two dimensions much larger than the third dimension. They retained their shape during annealing but changed size as the annealing time increased. The annealing time also affected the number of crystals, which increased with time. The layer was inhomogeneous with a random distribution of crystals.

Elemental analysis showed an increasing chromium tendency with decreasing acceleration voltage (Figure 2), which increased at the expense of iron and nickel. The increased chromium was also seen in the time dependence when the surface layer was richer in chromium for longer annealing times.

Figure 2 (EDS measurements) indicates that at 20 kV, the ratio of elements in the sample stabilized, which means that at this depth (1.3 µm), the average amounts of elements correspond to the original steel. This depth includes volumes with increased chromium and volumes with reduced chromium. We assume that the actual surface layer corresponds to approximately half of this depth, i.e., approximately 0.6 µm. Interestingly, when 30 kV was applied, the amount of iron decreased slightly, followed by a slight increase in molybdenum. These changes are within the measurement uncertainty, but they were observed this way for all eight samples. However, because this tendency was also observed in the unannealed sample, this fluctuation was considered a measurement error.

An inhomogeneous approx. 0.6 µm surface layer formed by chromium oxide was also manifested in the Mössbauer spectra. CXMS identified a decrease in effect between unannealed and annealed samples (Figure 6). This decrease corresponds to the attenuation of the secondary X-ray radiation by a homogeneous chromium layer with a thickness of approximately 0.1 µm. However, this decrease could also be caused by an inhomogeneous layer of greater thickness, corresponding to the inhomogeneous layer identified with an estimated thickness of 0.6 µm.

Since two methods, namely CEMS and CXMS, were used, it was possible to compare the phase composition of the samples at different depths corresponding to the penetration depth of electrons (0.3 µm) and X-ray radiation (10 µm). In the case of the CXMS measurements (Figure 5), the presence of a new phase was not detected; therefore, structural changes did not occur at this depth. On the other hand, in the case of CEMS, new phases appeared.

In samples annealed for ≥4 h, an austenitic–ferritic transformation took place (Figure 7 and Figure 8). This transformation was previously observed by CEMS [5] and showed austenitic stainless steel in a 0.3 μm surface layer transformed into duplex stainless steel by annealing at 550 °C. In addition to the austenitic–ferritic transformation, an Fe^3+^ doublet, which may correspond to iron oxide, or mixed Fe–Cr oxide, was identified in the surface layer by CEMS. XRD showed a mixed Fe–Cr oxide (Figure 8). The appearance of the doublet corresponding to the mixed Fe–Cr oxide proves the absence of an effective magnetic field; otherwise, a six-line split should appear [23]. Consequently, either the resulting product corresponding to the doublet in the Mössbauer spectroscopy has no magnetic order at room temperature or the product behaves superparamagnetically. It is well known that the critical particle size of the appearance of superparamagnetism of relevant oxidic iron compounds is 6–16 nm [24]. However, relatively narrow diffraction lines of Fe–Cr mixed oxide were identified in the XRD patterns (Figure 8, annealing for two hours). In the case of small particles or their clusters, these lines would be strongly broadened or not visible at all. Therefore, superparamagnetic relaxation is assumed to be negligible. The investigated oxide should be some mixed oxide of iron and chromium, which causes the appearance of only one doublet instead of a sextet [25]. Mixed Fe–Cr oxides are sensitive to non-equilibrium cation distribution, as demonstrated by different annealing procedures [26]. For example, in work [27] (Fe_1-x_Cr_x_)_2_O_3_ oxides were synthesized for 0 ≤ x ≤ 1 and doublets with IS = 0.33–0.36 mm/s and QS = 0.5–0.7 mm/s were observed in the Mössbauer spectrum. The parameters of the Fe^3+^ doublet identified in CEMS are close to the values reported in [27]. Mixed Fe–Cr oxides have been identified in stainless steel oxidation by XRD by other authors, for example, [1,25]. Therefore, we state that the oxidation product of the stainless steel surface we identified is a mixed Fe–Cr oxide.

## 5. Conclusions

The effect of temperature on the surface layer of stainless steel samples was studied. Annealing was carried out in air at a temperature of 550 °C during different periods of time. The study of the surface of steel parts proved that annealing resulted in the formation of an inhomogenous surface layer that prevents deep oxidation. The study of surface morphology showed the presence of leaf-shaped crystals of chromium oxide. The number and size of crystals increased with increasing annealing time, but the depth of the highly chromium-enriched surface layer was estimated to be 0.6 μm. Iron oxidized to a mixed Fe–Cr oxide, which was already formed during annealing for 2 h. At annealing times longer than 4 h, transformation of the austenitic to ferritic phase occurred, the degree of which increased with annealing time. All of these new phases were observed at approximately 0.3 µm depth. The occurrence of new phases was confirmed using CEMS, CXMS and XRD methods.

During stainless steel annealing at 550 °C, three transformation processes appeared in the surface layer, the intensity of which varied with time. Although we would not expect iron oxide to occur since stainless steel was annealed, oxidation occurred in a short time to form a mixed Fe–Cr oxide. At the same time, chromium diffused to the surface, and as soon as a certain amount of chromium was reached in the surface layer, chromium oxidation began to dominate. Chromium oxide crystals were formed, the amount of which increased on the surface and prevented deep oxidation. The protective layer of chromium oxide is thus only the next step after the formation of the mixed oxide. At the same time, the austenitic phase transformed into the ferritic phase. As a result, the austenitic iron transformed into a mixed Fe–Cr oxide and ferritic phase, which changed the properties of the stainless steel surface. Formed iron structures and chromium oxide must be taken into consideration for subsequent nanotechnological surface treatment.

## Figures and Tables

**Figure 1 materials-15-08871-f001:**
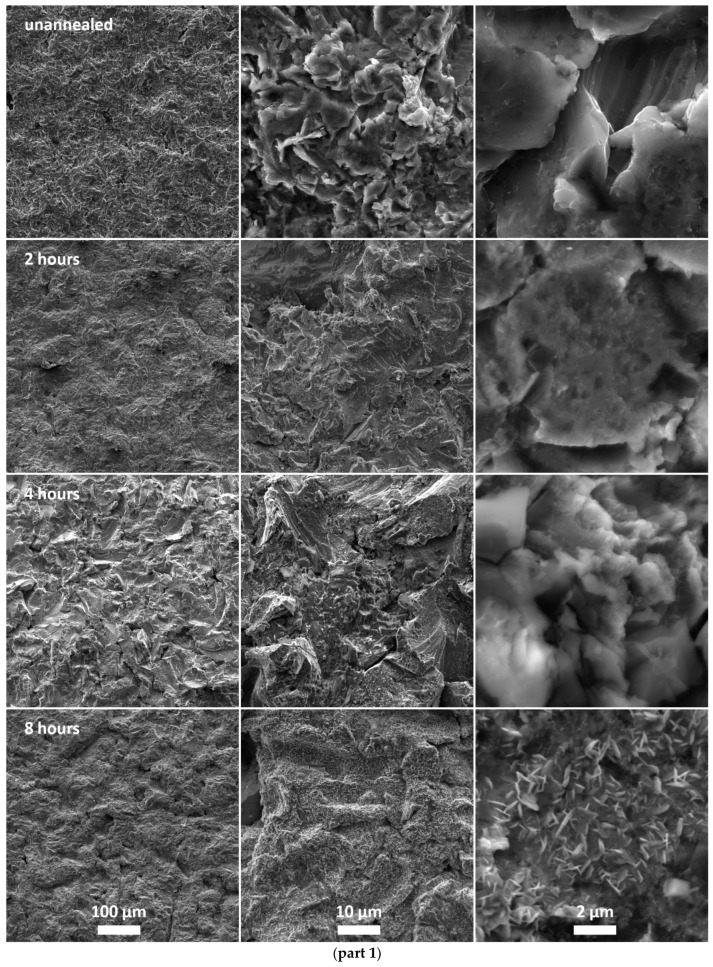
Scanning electron microscopy images at 1000 × (left), 10,000 × (middle) and 50,000 × (right) magnification: **part 1**—unannealed sample and samples annealed for 2, 4 and 8 h; **part 2**—samples annealed for 16, 32, 64 and 128 h.

**Figure 2 materials-15-08871-f002:**
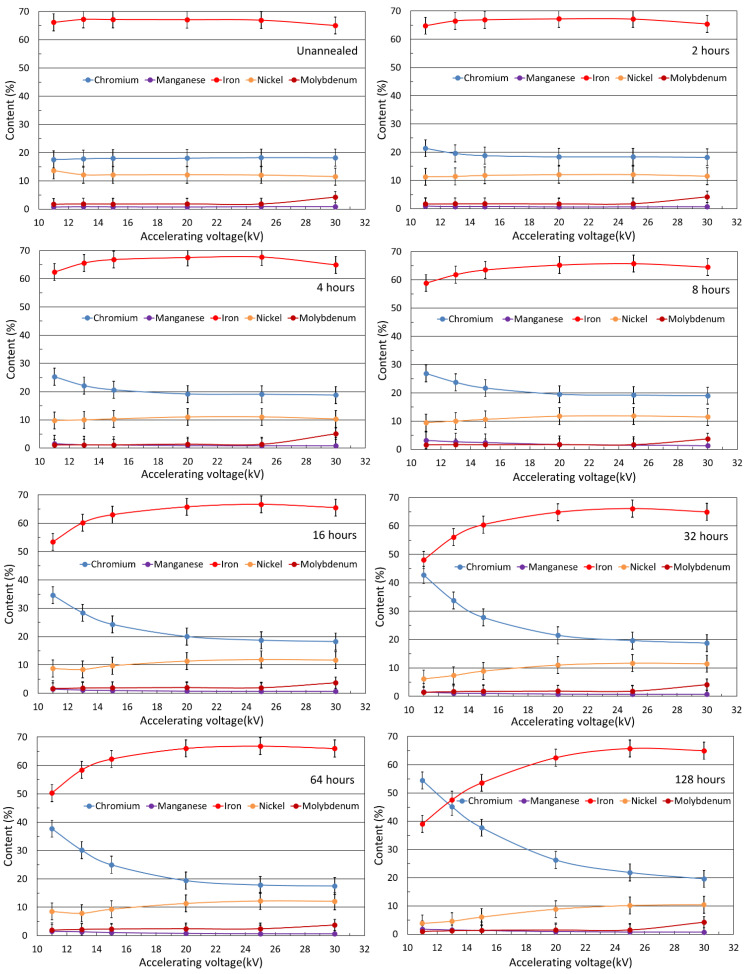
In-depth elemental analysis of selected samples by energy-dispersive X-ray spectroscopy.

**Figure 3 materials-15-08871-f003:**
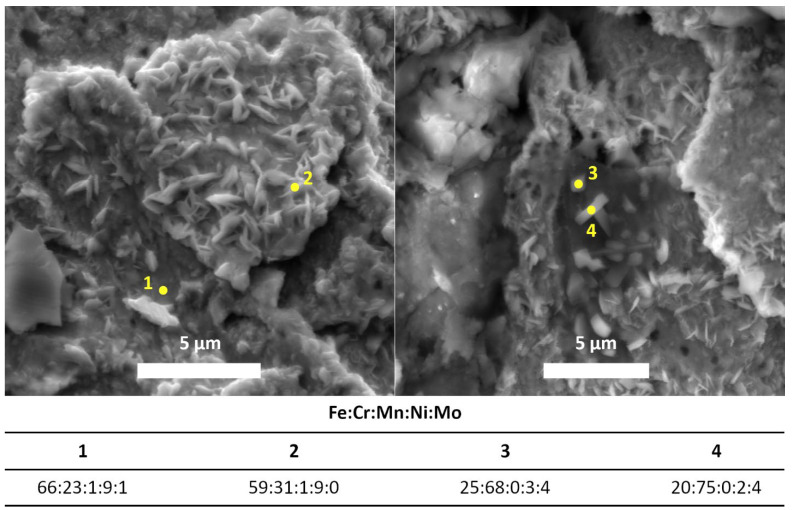
Elemental analysis at selected points of the sample annealed for 32 h at an accelerating voltage of 15 kV.

**Figure 4 materials-15-08871-f004:**
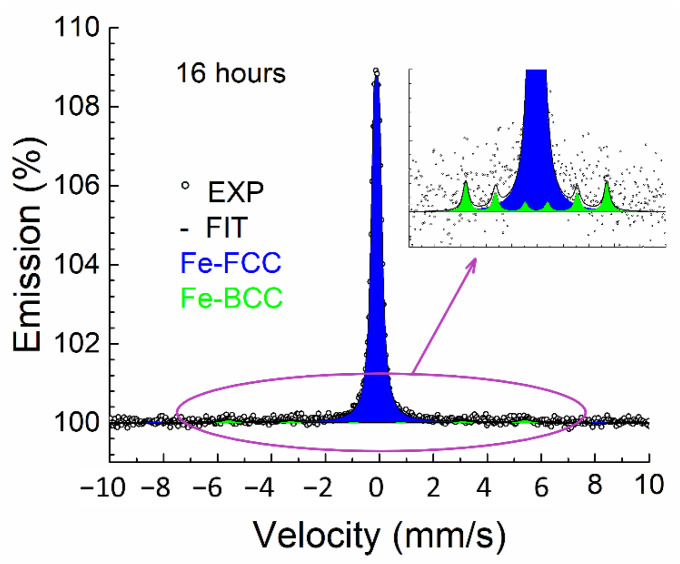
Conversion X-ray Mössbauer spectra of sample annealed at 550 °C for 16 h (measured for 430 h).

**Figure 5 materials-15-08871-f005:**
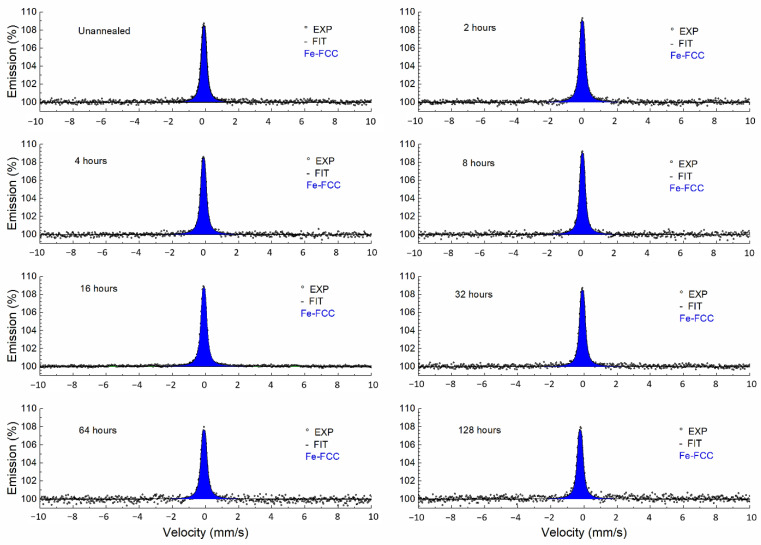
Conversion X-ray Mössbauer spectra of annealed samples.

**Figure 6 materials-15-08871-f006:**
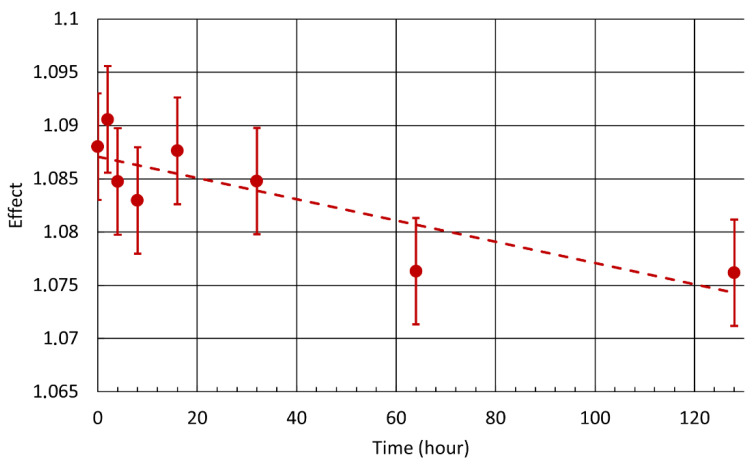
Effect in conversion X-ray Mössbauer spectroscopy in samples annealed at 550 °C.

**Figure 7 materials-15-08871-f007:**
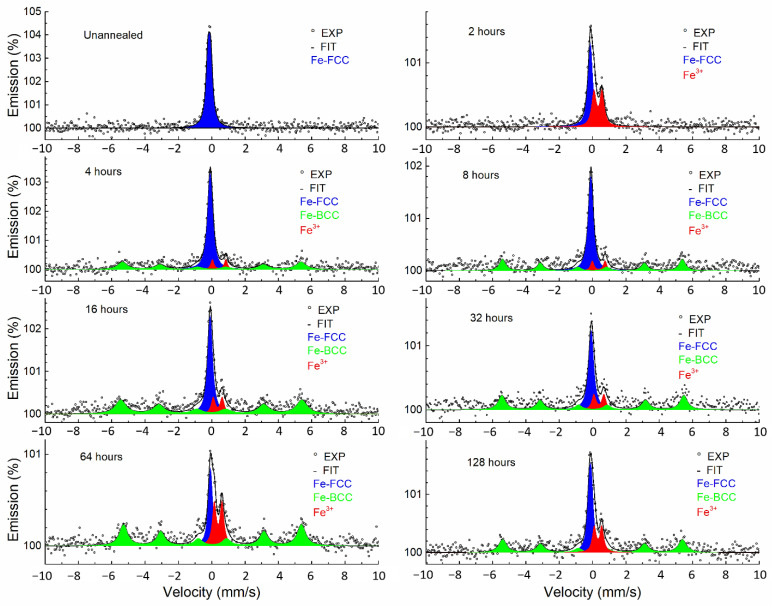
Conversion electron Mössbauer spectra of annealed samples.

**Figure 8 materials-15-08871-f008:**
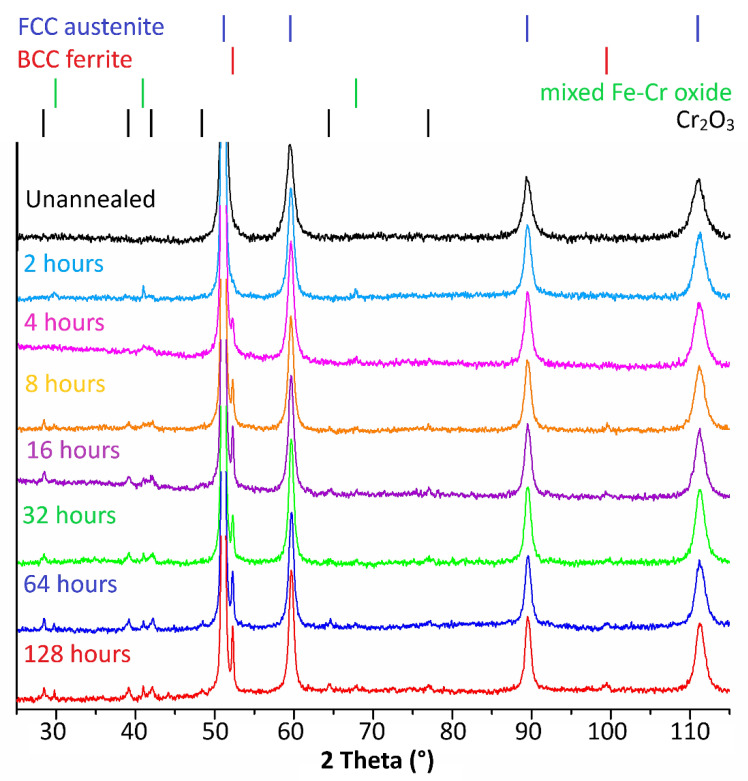
X-ray diffraction patterns of annealed samples.

**Table 1 materials-15-08871-t001:** Chemical composition of CL20ES steel powder (material data Concept Laser GmbH).

Fe	C	Si	Mn	P	S	Cr	Mo	Ni
Bal	≤0.030	0–1.0	0–2.0	≤0.045	≤0.030	16.5–18.5	2.0–2.5	10.0–13.0

**Table 2 materials-15-08871-t002:** Conversion X-ray Mössbauer spectroscopy parameters of all samples (IS, isomer shift; QS, quadrupole splitting; FWHM, full width at half maximum; B, hyperfine magnetic field; A, spectrum area).

Time (Hour)	Phase	IS (mm/s)	QS (mm/s)	FWHM (mm/s)	B (T)	A (%)
Unannealed	FCC austenite	−0.10 ± 0.01	0.17 ± 0.01	0.29 ± 0.01	-	100
2	FCC austenite	−0.08 ± 0.01	0.16 ± 0.01	0.28 ± 0.01	-	100
4	FCC austenite	−0.12 ± 0.01	0.16 ± 0.01	0.28 ± 0.01	-	100
8	FCC austenite	−0.10 ± 0.01	0.16 ± 0.01	0.28 ± 0.01	-	100
16	FCC austenite	−0.10 ± 0.01	0.16 ± 0.01	0.28 ± 0.01	-	95 ± 2
	BCC ferrite	−0.09 ± 0.05	-	0.52 ± 0.20	34.2 ± 0.5	5 ± 2
32	FCC austenite	−0.10 ± 0.01	0.16 ± 0.01	0.29 ± 0.01	-	100
64	FCC austenite	−0.07 ± 0.01	0.17 ± 0.01	0.29 ± 0.01	-	100
128	FCC austenite	−0.10 ± 0.01	0.17 ± 0.01	0.28 ± 0.01	-	100

**Table 3 materials-15-08871-t003:** Conversion electron Mössbauer spectroscopy parameters of all samples (IS, isomer shift; QS, quadrupole splitting, FWHM, full width at half maximum; B, hyperfine magnetic field; A, spectrum area).

Time (Hour)	Phase	IS (mm/s)	QS (mm/s)	FWHM (mm/s)	B (T)	A (%)
Unannealed	FCC austenite	−0.12 ± 0.01	0.15 ± 0.01	0.25 ± 0.01	-	100
2	FCC austenite	−0.13 ± 0.01	0.10 ± 0.03	0.28 ± 0.03	-	54 ± 2
	Fe^3+^ doublet	0.33 ± 0.01	0.49 ± 0.02	0.33 ± 0.03	-	46 ± 2
4	FCC austenite	−0.09 ± 0.01	0.13 ± 0.01	0.28 *	-	54 ± 2
	BCC ferrite	0.03 ± 0.03	-	0.79 ± 0.09	33.1 ± 0.5	40 ± 2
	Fe^3+^ doublet	0.36 *	0.52 *	0.28 *	-	6 ± 2
8	FCC austenite	−0.09 ± 0.01	0.12 ± 0.01	0.28 *	-	57 ± 2
	BCC ferrite	0.01 ± 0.02	-	0.42 ± 0.06	33.6 ± 0.5	36 ± 2
	Fe^3+^ doublet	0.36 *	0.52 *	0.28 *	-	7 ± 2
16	FCC austenite	−0.10 ± 0.01	0.12 ± 0.01	0.28 *	-	37 ± 2
	BCC ferrite	−0.02 ± 0.03	-	0.77 ± 0.07	33.8 ± 0.5	51 ± 2
	Fe^3+^ doublet	0.38 ± 0.02	0.51 ± 0.03	0.28 *	-	12 ± 2
32	FCC austenite	−0.09 ± 0.01	0.11 ± 0.02	0.28 *	-	40 ± 2
	BCC ferrite	0.02 ± 0.02	-	0.50 ± 0.06	33.9 ± 0.5	46 ± 2
	Fe^3+^ doublet	0.39 ± 0.02	0.57 ± 0.03	0.28 *	-	14 ± 2
64	FCC austenite	−0.08 ± 0.01	0.07 ± 0.03	0.28 *	-	24 ± 2
	BCC ferrite	0.04 ± 0.02	-	0.58 ± 0.06	33.2 ± 0.5	52 ± 2
	Fe^3+^ doublet	0.38 ± 0.01	0.48 ± 0.02	0.28 *	-	24 ± 2
128	FCC austenite	−0.10 ± 0.01	0.12 ± 0.02	0.28 *	-	41 ± 2
	BCC ferrite	0.02 ± 0.02	-	0.56 ± 0.08	33.6 ± 0.5	40 ± 2
	Fe^3+^ doublet	0.35 ± 0.02	0.46 ± 0.03	0.28 *	-	19 ± 2

* fixed parameter.

## Data Availability

The data presented in this study are available on request from the corresponding author.

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
