# Peer review of "Phase Changes in the Surface Layer of Stainless Steel Annealed at a Temperature of 550 °C"

_materials, 2022, doi:10.3390/ma15248871_

Round 1

Reviewer 1 Report

I suggest rejecting this paper to be published on materials The reasons are as follows:

1. The content of this paper is flawed. The description of the experimental results is not deep enough. It is just a simple description of the results, lacking a deeper scientific explanation

2. The innovation of this paper is insufficient, and the research entry point is single

3. There are defects in the arrangement of article structure writing. There are too many experimental steps and methods in the description, and the interpretation of experimental results is too little and not deep enough. This is more like an experimental report type article than a research article.  

Reviewer 2 Report

1. The grammatical and typo errors must be revised.

2. Authors annealed the stainless steel samples at 2, 4, 8, 16, 32, 64 and 128 hours, but they depicted the SEM of only 0, 2, 4, 8 and 16 hours. It is recommended to show SEM of all the samples.

3. In few places, spacing between the words is not uniform.

4. Figure 4, looks cropped, it is recommended to add the full figure.

5. Instead of showing the figure 8. XRD of all the samples individually, it is recommended to over plot them in a single image for easy comparision.

6. It seems, the font size and type of references is not according to the journal requirement. Revise them.

Reviewer 3 Report

This work mainly studied the surface morphology, elemental and phase composition of CL20ES steel annealed at 550 by means of several testing tools, e.g., CEMS, EDS, XRD and SEM. An austenite-ferrite phase transformation was detected and characterized. However, the paper seems to focus on the difference of testing methods, rather than the research aim of phase changes of stainless steel as shown in the tittle. Thus, the arrangement of the present version of manuscript is confused. Major revision is needed to fulfill the high requirements of the journal. Below you can find the comments:

 (1) In sections of abstract and conclusions, the obtained results should be described clearly and precisely. The present version gives more description on testing tools, with weak conclusions on phase changes of stainless steel annealed at 550 .

(2) In introduction, as reviewed by authors, 316L stainless steel has been studied by many researchers, some phase transformations were observed. so how about CL20ES steel? What’s the novelty of present study? It’s unclear.

(3) In Fig.1, the SEM images show well the difference in morphology of the steels annealed at different holding time, such as the needle-like compounds shown in Fig.1. But the descriptions and discussions on characteristics of these surfaces are less and insufficient. The same problems can be found in Figs.2-4.

(4) In Fig.2, there is a big difference in chromium content when holding time increases from 16 to 128h, while the iron content seems to have no change. Please give the further reason.

(5) In section 4, the variations of microstructure and composition of steels should be clearly stated with increasing holding time. However, the authors compare the different testing methods.

(6) The conclusion is very weak, probably having little relationship with the paper tittle.

Round 2

Reviewer 1 Report

1. The authors' description of the current research situation is too old and does not fully describe the current research situation.

2. Only one of the 23 references has been published in the past 5 years. Please give a brief introduction to the research achievements in recent years.

Author Response

The introduction chapter has been expanded with the latest citations. Also, part of the reference was modified.

Reviewer 3 Report

The conclusion seems to be not "conclusion". It is a description on some figures, such as "During stainless steel annealing at 550 °C, three transformation processes appear in the surface layer, the intensity of which varies with time. Oxidation occurs in a short time to form a mixed Fe–Cr oxide.  At the same time, chromium diffuses to the surface, and as soon as a certain amount of chromium is reached in the surface layer, chromium oxidation begins to dominate. Chromium oxide crystals are formed, the amount of which increases on the surface and prevents deep oxidation. At the same time, the austenitic phase transforms into the ferritic phase."

This should be improved further in the revised version.

Author Response

The conclusion chapter has been modified and expanded.